# Alanine-Dependent TCA Cycle Promotion Restores the Zhongshengmycin-Susceptibility in *Xanthomonas oryzae*

**DOI:** 10.3390/ijms24033004

**Published:** 2023-02-03

**Authors:** Zhenyu Zou, Meiyun Lin, Peihua Shen, Yi Guan

**Affiliations:** Fujian Key Laboratory of Marine Enzyme Engineering, College of Biological Science and Engineering, Fuzhou University, Fuzhou 350116, China

**Keywords:** *Xanthomonas oryzae*, Zhongshengmycin, antimicrobials, metabolomics, TCA cycle, alanine

## Abstract

*Xanthomonas oryzae* pv. oryzicola (Xoo) is a plant pathogenic bacterium that can cause rice bacterial blight disease, which results in a severe reduction in rice production. Antimicrobial-dependent microbial controlling is a useful way to control the spread and outbreak of plant pathogenic bacteria. However, the abuse and long-term use of antimicrobials also cause microbial antimicrobial resistance. As far as known, the mechanism of antimicrobial resistance in agricultural plant pathogenic bacteria still lacks prospecting. In this study, we explore the mechanism of Zhongshengmycin (ZSM)-resistance in Xoo by GC-MS-based metabolomic analysis. The results showed that the down-regulation of the TCA cycle was characteristic of antimicrobial resistance in Xoo, which was further demonstrated by the reduction of activity and gene expression levels of key enzymes in the TCA cycle. Furthermore, alanine was proven to reverse the ZSM resistance in Xoo by accelerating the TCA cycle in vivo. Our results are essential for understanding the mechanisms of ZSM resistance in Xoo and may provide new strategies for controlling this agricultural plant pathogen at the metabolic level.

## 1. Introduction

Natural catastrophes, pests, and plant pathogens threaten the production of food crops worldwide, with diseases caused by plant pathogens accounting for 16% of global crop yield losses [1,2]. *Xanthomonas oryzae* pv. oryzicola (Xoo) is a rod-shaped Gram-negative bacterium that can infect rice leaves through wounds or stomata, thereby causing bacterial blight (BB), one of the most devastating diseases of rice [3]. Low concentrations of antimicrobials can achieve excellent bactericidal levels and have negligible toxicity to plants [4]. Presently, various antimicrobials are used to control BB, such as streptomycin, oxytetracycline, neomycin, and Zhongshengmycin [4,5,6,7].

Zhongshengmycin (ZSM herein) is a type of N-glucosides agricultural antimicrobial developed by the Chinese Academy of Agricultural Sciences Institute of Biological Control, which works by interrupting the synthesis of protein peptide bonds of pathogenic microorganisms to inhibit the growth of bacteria and fungal mycelium and prevents the germination of spores [8]. ZSM has significant effects on the control of rice bacterial blight and walnut bacterial blight, and it also has a good controlling effect on *Nilaparvata lugens* [5,9,10]. However, the misuse and long-term use of antimicrobials have also induced a new common problem—microbial antimicrobial resistance [11]. For example, Xu and Zhu collected Xoo that showed resistance to streptomycin in the field during previous decades [12,13]. In recent years, shenazinomycin-resistant Xoo was also screened [14]. Therefore, studying and unveiling the mechanisms of microbial antimicrobial resistance is vital to solve this problem.

The metabolome examines the overall metabolic level of a biological system under specific conditions, and it is the chemical entity transformed in the metabolic process of a physical system [15]. It can monitor bacterial changes in response to antimicrobials and provide comprehensive metabolic information, which can aid understanding of the metabolic processes behind antimicrobial resistance mechanisms, physiological and biochemical states [16,17,18]. Recent studies showed that increased spermine and L-ascorbate are associated with the acquisition of ofloxacin resistance in *Escherichia coli* [19]; triclosan inhibits the biosynthesis of fatty acids to restore the sensitivity of *Vibrio alginolyticus* to ceftazidime [20]; and alleviation of fitness cost in competition between susceptible and resistant *E. coli* at sub-MIC doxycycline [21]. Hence, metabolomic analysis has been widely applied in investigating microbial antimicrobial resistance and is expected to provide a new solution for reversing bacterial resistance.

This study used GC-MS-based metabolomics to characterize the metabolomic changes in ZSM-sensitive/resistant Xoo strains (Z173-S and Z173-R_ZS_, respectively). We found that down-regulation of the TCA cycle may be the critical metabolic characteristic involved in the resistance to ZSM in Z173-R_ZS_. In addition, exogenous alanine enhances the controlling effect of Xoo against ZSM by promoting the TCA cycle. Our results are important for understanding the mechanism of resistance to ZSM in Xoo and provide a novel approach to restoring the sensitivity of drug-resistant Xoo to ZSM.

## 2. Results

### 2.1. Basic Phenotypes of Z173-S and Z173-R_ZS_

To understand the resistance mechanism of Xoo to ZSM, we measured the growth curve of the sensitive strain Z173-S, and successively passaged it with ZSM to obtain a ZSM-resistant strain, Z173-R_ZS_. After measurement, we found that Z173-S was in the growth retardation phase from 0 to 6 h, entered the logarithmic growth phase after 6 h, and reached the late logarithmic growth phase after 18 h of culture (Figure 1A). Therefore, 18 h was determined to be the best fermentation time for 24-deep-well plates. The ZSM-sensitive/resistant strains of Xoo, Z173-S and Z173-Rzs, were stored in our lab. The minimum inhibitory concentrations (MIC) of ZSM against Xoo were determined by the 24-deep-well plates according to the method described previously [22]. Our results showed that the MIC of Z173-Rzs was 48 μg/mL, which was 16-fold higher than that of the susceptible strain Z173-S (Figure 1B).

### 2.2. Metabolomic Analysis of Z173-S and Z173-R_ZS_

The metabolomic spectra of Z173-R_ZS_ and Z173-S were obtained and analyzed by the GC-MS-based metabolomics approach to explore the metabolic difference between the two strains. Essentially, Z173-S and Z173-R_ZS_ were set as the control or treatment groups containing 5 biological replicates, respectively, resulting in a total of 10 datasets in the metabolomic profile. As a result, ninety-one metabolites were identified. The representative ion current chromatograms of the two samples are shown in Appendix A. All of the metabolites were searched through KEGG and HMDB to determine their biological categories. The result showed that these 91 metabolites could be divided into six major categories, i.e., carbohydrates (10.99%), amino acids (14.29%), lipids (30.77%), nucleotides (2.20%), carboxylic acids (10.99%), and others (30.77%) (Appendix A).

Among the 91 identified metabolites, a total of 80 differential metabolites (DMs) were detected through the Mann-Whitney U Test by SPSS analysis software, with the asymptotically significant *p* < 0.05, and their metabolic abundances were visualized by differential metabolite heat maps (Figure 2A). The Z-Scores of these differential metabolites range from −20 to 50. Specifically, compared to Z173-S, 46 metabolites were increased, and 34 metabolites were decreased in Z173-R_ZS_ (Figure 2B). The proportions of these differential metabolites could be categorized into carbohydrates (11.25%), amino acids (12.50%), lipids (32.50%), nucleotides (2.50%), carboxylic acids (8.75%), and other (32.50%) (Figure 2C). The number of the up- or down-regulated DMs in each category is shown in Figure 2D. The above results indicate that Z173-S and Z173-R_ZS_ have different metabolic profiles, which may be related to the ZSM resistance in Z173-R_ZS_.

### 2.3. Enrichment Pathway Analysis

Through the Metaboanalyst website (http://www.MetaboAnalyst.ca (accessed on 18 October 2022)), we further analyzed the metabolic differences between Z173-S and Z173-Rzs. In total, 80 DMs were enriched in 41 metabolic pathways, and the 8 key metabolic pathways were identified as significantly involved pathways with a *p* < 0.05 (Figure 3A), i.e., the glutathione metabolism, the alanine, aspartate and glutamate metabolism, the glyoxylate and dicarboxylate metabolism, the arginine and proline metabolism, the aminoacyl-tRNA biosynthesis, the citrate cycle (TCA cycle), the arginine biosynthesis, and the taurine and hypotaurine metabolism. The relative abundances of the differential metabolites in Z173-R_ZS_ and Z173-S are shown in Figure 3B. These results suggest that the metabolic state can change significantly as the level of Xoo resistance to ZSM changes.

### 2.4. The Depressed TCA Cycle Is Associated with Z173-R_ZS_ Resistance to ZSM

Among the eight significantly involved pathways in Z173-Rzs, the TCA cycle could be associated with many other pathways such as the alanine, aspartate, and glutamate metabolism, the glyoxylate and dicarboxylate metabolism, the arginine and proline metabolism, the arginine biosynthesis, and the glutathione metabolism. Therefore, the TCA cycle is hypothesized to be one of the key metabolic pathways involved in Z173-R_ZS_ resistance to ZSM.

The hypothesis was verified by enzyme activity assay and related gene expression levels of three critical enzymes in the TCA cycle. Specifically, the activities of succinate dehydrogenase (SDH), α-ketoglutarate dehydrogenase (α-KGDH), and pyruvate dehydrogenase (PDH) were all significantly decreased in Z173-R_ZS_ compared with Z173-S (Figure 4A). In addition, the expression levels of most genes encoding these three enzymes were also significantly down-regulated in Z173-R_ZS_ (Figure 4B). These results indicate that the TCA cycle in Z173-R_ZS_ was depressed.

### 2.5. Inhibition of the TCA Cycle Promoted the Survival Rate of Z173-R_ZS_ under ZSM Stress

As the TCA cycle has been proven to be depressed in Z173-R_ZS_, we further determine whether the inhibition of the TCA cycle is beneficial to ZSM resistance in the Xoo. Therefore, an inhibitor of PDH, furfural, was used to suppress the TCA cycle of Z173-R_ZS_. Our results show that the addition of furfural caused the significant enzyme activity inhibition of PDH (Figure 5A). Moreover, with the 0.17 μL/mL furfural addition, the survival rate of Z173-R_ZS_ under 6 μg/mL ZSM was significantly increased by 5.14% when compared to that of the Z173-R_ZS_ without furfural (Figure 5B). These results suggest that the TCA cycle in Z173-R_ZS_ is suppressed by adding the PDH inhibitor, and the inhibition of the TCA cycle positively affects the ZSM resistance in Xoo.

### 2.6. Exogenous Alanine Elevates the Sensitivity of Z173-R_ZS_ to ZSM

Alanine was proven to be associated with microbial antimicrobial resistance in previous studies [23,24,25]. In the present study, alanine is significantly down-regulated in Z173-Rzs. Therefore, the effect of exogenous alanine in the TCA cycle and the ZSM resistance in Z173-Rzs was detected. The experimental results show that when Z173-R_ZS_ was incubated with 0.04 mol/L alanine, the activity of α-KGDH and SDH in the TCA cycle was significantly increased, and the survival rate of Z173-R_ZS_ decreased by 16.89% (Figure 6A,B). These results reveal that exogenous alanine could promote the TCA cycle and enhance the controlling effect of ZSM on Xoo.

## 3. Discussion

*Xanthomonas oryzae* pv. oryzicola (Xoo) is a pathogen that causes *Xanthomonascampestrispv. oryzae*. With the preventive use of antimicrobials, the emergence of drug-resistant bacteria of Xoo has occurred. Restoring the sensitivity of drug-resistant bacteria to antimicrobial agents is crucial for the sustainable development of agriculture. Therefore, there is an urgent need to better understand the mechanism of antimicrobial resistance in resistant bacteria and develop new drug treatment strategies. In recent years, studying drug resistance from the perspective of metabolism has attracted attention and research. Aros-Calt used LC-MS to study the methicillin-resistant mechanism of *Staphylococcus aureus* [26]. The antioxidant properties of cysteine or cysteine-derived metabolites can contribute to the development of drug resistance in *Salmonella typhimurium* [27]. Later, on this basis, Peng proposed a new mechanism: using metabolites to make antimicrobials control antimicrobial-resistant pathogens to solve antimicrobial resistance [28]. Therefore, it is of great significance to study the metabolites to restore the sensitivity of drug-resistant bacteria. Our previous study compared and analyzed the metabolomic features of Xoo-S and Xoo-ZSM. The results suggest that a decrease in the P cycle is a characteristic feature of Xoo-ZSM and that exogenous alanine also promotes P cycling in ZSM-resistant strains. In this study, we employed different pathogenic strains, and the depressed TCA cycle was proven to be associated with ZSM resistance in the antimicrobial-resistant strain, Z173-Rzs, and alanine was found to be a potential substance to enhance the controlling effect of ZSM against Xoo.

Firstly, we used GC-MS-based metabolomic analysis to identify the intracellular metabolites in Z173-S and Z173-R_ZS_ and analyze the differential metabolic pathways related to antimicrobial resistance. Glutathione metabolism and glutamic acid, aspartate, and glutamic acid metabolism were most significantly affected by the KEGG pathway. At the same time, the abundance of alanine, citric acid, and oxalic acid also decreased significantly. Our study showed that the metabolite abundance of Z173-S and Z173-R_ZS_ are very different, and the TCA cycle pathway may be related to the ZSM antimicrobial in Z173-R_ZS_. These results were further verified by the fluctuation of the activity of three enzymes, i.e., SDH, PDH, and α-KGDH, in the TCA cycle and their corresponding gene transcription levels. The TCA cycle has been reported to be associated with bacterial antimicrobial resistance in previous work. For example, promoting the TCA cycle can increase the resistance to chloramphenicol in *Edwardsiella tarda* [29]. Inhibition of the TCA cycle could lead to the formation of *Staphylococcus aureus* with high drug resistance [30].

Secondly, the TCA cycle was inhibited by adding furfural, an inhibitor of pyruvate dehydrogenase, to test the relationship between the TCA cycle and the drug resistance in the ZSM-resistant strain, Z173-R_ZS_. Our work showed that with the inhibition of the TCA cycle, the ZSM-controlling effect was also reduced. Inhibition of the TCA cycle is associated with the acquisition of antimicrobial resistance by *Vibrio alginolyticus, Vibrio cholerae*, and *Pseudomonas aeruginosa* [31,32,33]. These results support the conclusion that the depression of the TCA cycle was associated with the ZSM antimicrobial resistance in the Xoo and are consistent with previous reports that down-regulation of the TCA cycle leads to the acquisition of bacterial resistance. Therefore, our work confirms the relationship between the TCA cycle and the antimicrobial resistance developing in the plant pathogenic bacterium Xoo. To our knowledge, this is the first report that reveals the relationship between the TCA cycle and ZSM-resistance development in the Xoo strain.

Recent studies have shown that some key metabolites play a regulatory role in antimicrobial-related bacterial control. For example, exogenous fructose and maltose alter the antimicrobial sensitivity of Lan-negative bacteria through metabolic regulation [34]. Exogenous citrulline and glutamine contribute to reversing the resistance of *Salmonella* to apramycin [35]. On the other hand, adding fructose can restore the sensitivity of *Edwardsiella tarda* to kanamycin [36]. In the present study, alanine was used to assay the effect of exogenous metabolite on antimicrobial-related bacterial controlling efficiency in the Xoo. As a result, the addition of alanine promoted the TCA cycle and enhanced the ZSM-mediated controlling efficiency in Z173-R_ZS_. Therefore, our work provides an applicable solution to controlling the ZSM-antimicrobial-resistant Xoo. Alanine could be used as a candidate in combination with ZSM, or as a synthetic precursor in developing new antimicrobials, for ZSM controlling in application. This study provides an important reference for the practical application of drugs against bacterial resistance and the development of new drugs. However, further experiments are needed to explore the specific mechanism of how the alanine functions in the TCA-associated ZSM-mediated controlling efficiency in Z173-R_ZS_.

## 4. Materials and Methods

### 4.1. Bacterial Strains and Growth Conditions

The antimicrobial-sensitive strain of Xoo (Z173-S) was provided by our laboratory. Through the subculture of Z173-S in PSA medium (peptone 10 g/L, sucrose 10 g/L, sodium glutamate 1 g/L, agar 2% for solid medium, pH 7.0) containing 1/2 MIC of ZSM, the strain Z173-R_ZS_ with resistance to ZSM was isolated. These two bacteria were both grown at 30 °C for incubation.

### 4.2. Measurement of Growth Curve

Z173-S initial cells from frozen stock were rapidly grown in PSA agar plates in a 30 °C incubator for static incubation for 2.5 d. A single colony was seeded in 3 mL of PSA broth medium at 200 rpm and 30 °C in a shaker overnight, and 1% (*v*/*v*) seed liquid was added in 24-deep-well plates with 2 mL of PSA liquid medium at 600 rpm and 30 °C shake cultivation. The biomass was valued at different time points (0.5 h, 1 h, 2 h, 4 h, 6 h, 8 h, 10 h, 12 h, 14 h, 16 h, 18 h, 20 h, 22 h, 24 h, and 28 h) as the OD_600_ using the Multiskan Sky microplate reader. Three biological replicates were set for each time point.

### 4.3. Minimum Inhibitory Concentration Detection (MIC)

The single clones were cultured overnight at 30 °C with rigorous shaking of the tube containing 3 mL of PSA medium, followed by transfer to 24-deep-well plates equipped with 2 mL of PSA broth medium with 2-fold serially diluted ZSM at 0, 30, 60, 120, 240, 480, 960, and 1920 μg/mL at 600 rpm, 30 °C for 18 h. The value of OD_600_ at the lowest antimicrobial concentration, which inhibits bacterial growth, was recorded as the minimum inhibitory concentration (MIC). All experiments were carried out with biological triplicates.

### 4.4. Metabolomic Profiling

The sample preparation procedure was previously described [22]. In brief, overnight cultures of Z173-S and Z173-R_ZS_ were seeded into 50 mL of PSA liquid medium at 30 °C, 200 rpm to OD_600_1.0. The Xoo thallus were collected and centrifuged at 4 °C and 10,000 rpm for 5 min, and the supernatant was discarded and washed three times with 0.9% saline. After that, 10 mL of saline and 20 mL of frozen methanol were added immediately into the sample and set for at least 1 h to terminate the metabolic process of the cells. Bacterial cells collected were resuspended with 500 μL of frozen methanol with 10 μL of ribitol at 0.2 mg/mL as an internal reference. Cells were then sonicated at power 120 W for 5 min and centrifuged at 4 °C at 12,000× *g* for 10 min, and 500 μL of supernatant was dried with a nitrogen blower. Samples were then silanized and derivatized with 80 μL (20 mg/mL) of methoxyamine hydrochloride and 80 μL of N-methyl-N-trimethylmethilane trifluoroacetamide (MSTFA, Sigma-Aldrich, USA) and incubated at 37 °C, 200 rpm for 30 min, respectively. The samples were centrifuged at 4 °C at 12,000× *g* for 5 min and then analyzed with a GC-MS system. Each sample had five biological repeats.

### 4.5. GC-MS Detection and Statistical Analysis

GC-MS detection was carried out using the two-stage technique as described previously [37]. Briefly, a 1 μL derivatized sample was injected into a DBS column (Agilent Technologies, 30 m × 250 μm × i.d. × 0.25 μm) using the splitless injection and was analyzed by Agilent G1701EA GC-MSD ChemStation (Agilent, Saint Louis, MO, USA). The following parameters were set for the GC-MS oven: initial temperature programmed, 85 °C; hold, 3 min; heating rate, 5 °C/min; heating temperature, 285 °C; final temperature, 310 °C; hold, 7 min; and heating rate, 20 °C/min. Electron impact ionization (EI) mode was selected, and ionization was of 70 eV energy. Helium was used as the carrier gas with a flow rate of 1 mL/min. The range of mass full scan mode was 50−600 m/z. All compounds were identified using the chemical analysis website NIST Chemistry WebBook (https://webbook.nist.gov/chemistry/ (accessed on 9 August 2022). For enrichment pathway analysis, the MetaboboAnalyst online website (www.Metaboanalyst.ca (accessed on 18 October 2022)) was used to determine the differential metabolic pathways involving the metabolites. Excel 2013 was used for data filtering and merge operations. The significant differences between metabolites in the control and experimental groups were compared using the Mann-Whitney U Test (*p* < 0.05) with SPSS 19.0 (IBM, New York, NY, USA).

### 4.6. Antimicrobial Bactericidal Assays

The bacterial cells of Z173-R_ZS_ were cultured overnight and collected by centrifugation for 5 min at 8000 rpm, and the original nutrients were washed away with 0.9% normal saline of 30 mL. In the M9 minimum medium with 10.0 mM sodium acetate, 2.0 mM magnesium sulfate, and 0.1 mM calcium chloride, the samples were resuspended to 0.2 of OD_600_. In order to determine the effect of exogenous additions on Z173-R_ZS_ cells, the concentrations of alanine were set as 0, 0.01, 0.02, and 0.04 mol/L, and the concentration gradient of furfural was set to 0, 0.0425, 0.085, and 0.17 μL/mL. After incubation at 30 °C and 200 rpm for 6 h, the survival rate of Z173-R_ZS_ incubated at 30 °C for 36–48 h was calculated by continuous gradient dilution, and the CFU was counted. The survival rate was determined by dividing the CFU obtained from a treated sample by the CFU obtained from the control.

### 4.7. qRT-PCR

Quantitative real-time polymerase chain reaction (qRT-PCR) was used to detect the expression level of enzymes related to the TCA cycle according to the method described above [11]. In short, after the strain was cultured to 0.6 of OD_600_, the 2 mL of liquid was collected and transferred to a new enzyme-free centrifuge tube. In the next step, the total RNA was extracted according to the total RNA extraction kit, and its concentration and purity were determined by full-wavelength enzyme labeling μDropTablet (Thermo, Massachusetts, USA) then reverse transcribed into cDNA by one-step synthesis (TransGen, Beijing, China). The QRT-PCR reaction was carried out on a 96-well plate of the fluorescence quantitative PCR instrument (Thermo, Waltham, MA, USA). The primers are listed in Appendix A. Each well contained a total volume of 20 μL of liquid including 2 μL cDNA template, 0.4 μL each pair of primers, 10 μL 2 × Perfecttart Green qPCR SuperMix, 0.4 μL 50 × Passive Reference Dye optional, and 6.8 μL PCR-grade water. The relative expression of each gene was calculated by the 2^−ΔΔCt^ method. At least three biological replicates were carried out.

### 4.8. Enzyme Activity Determination

The activities of three key enzymes—succinate dehydrogenase (SDH), pyruvate dehydrogenase (PDH), and α-ketoglutaric dehydrogenase (α-KGDH)—were carried out according to the instructions of the commercial enzyme activity detection kits (Solabio, Beijing, China), which were calculated by the protein concentration determined by BCA quantitative determination kit (Meilun, Dalian, China). The incubation of Xoo bacteria was performed as mentioned above. In short, exogenous additions (alanine, furfural) were added separately to the PSA medium overnight to centrifugate at 4 °C, 8000 rpm for 5 min. Before determination, the bacterial cells were fragmented with an ultrasonic crusher (intensity 60%, open cycle 5 s, closed cycle 5 s, time 6 min). The supernatants were collected by centrifugation at 11,000× *g* for 10 min at 4 °C. The dehydrogenase activity was assayed in triplicate.

## 5. Conclusions

In summary, we demonstrated that the metabolite abundance of susceptible and resistant bacteria is significantly different and that specific biomarkers are associated with ZSM resistance. In contrast to our previous studies, the mechanism of ZSM resistance in Xoo was further revealed. Our work confirmed that down-regulation of the TCA cycle is a crucial metabolic feature of Z173-R_ZS_ and a positive effect of alanine in reversing antimicrobial resistance was identified. In addition, we proposed an applicable candidate, alanine, to be used to enhance the antimicrobial-mediated Z173-R_ZS_ controlling efficiency. In subsequent studies, we will consider reprogramming functional metabolic pathways by metabolic regulators or combining other omics, including transcriptomics, whole genome sequencing, and proteomics, to explore the resistance mechanism of drug-resistant Xoo through metabolism, transcription, gene, and protein levels.

## Figures and Tables

**Figure 1 ijms-24-03004-f001:**
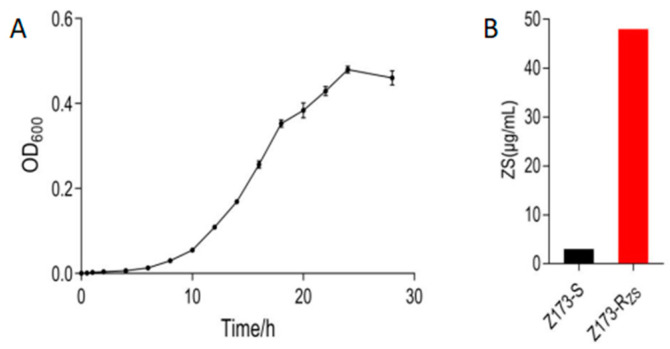
Phenotype of Z173-S and Z173-R_ZS_. (**A**) The growth curve of Z173-S. (**B**) The minimum inhibitory concentration of Z173-S and Z173-R_ZS_. Results are displayed as mean ± SD. At least three biological repeats were carried out.

**Figure 2 ijms-24-03004-f002:**
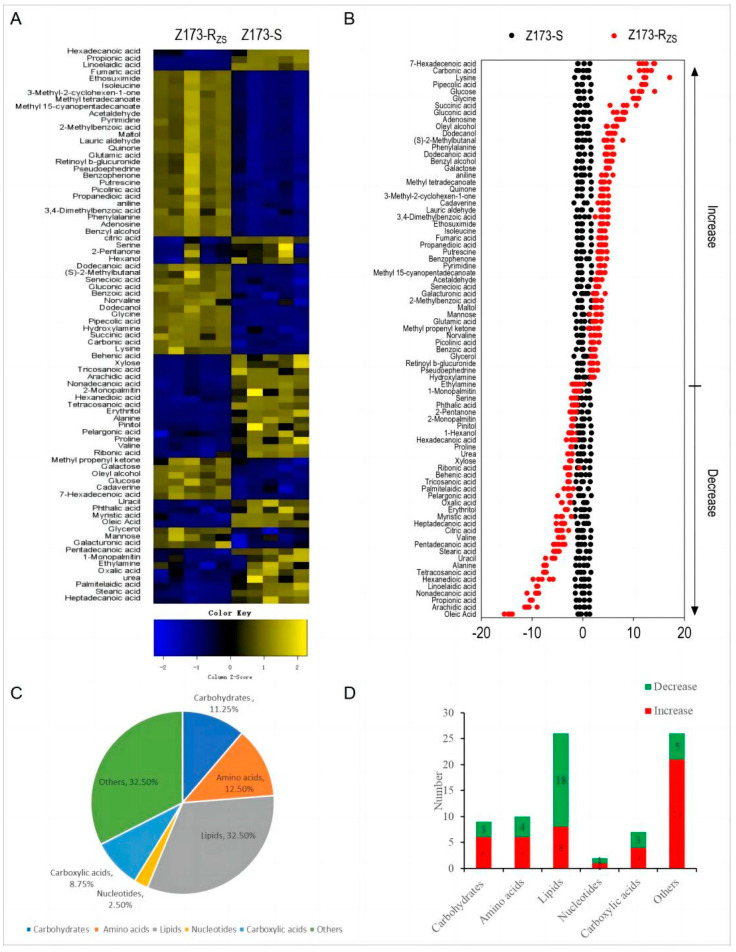
Identification of differential metabolites in Z173-S and Z173-R_ZS_. (**A**) Heat map of unsupervised hierarchical clustering of varied abundance of metabolites. Yellow and blue indicate increase and decrease of metabolites relative to the median metabolite level, respectively (see color scale). (**B**) Z-score plots. Each point represents one metabolite in one technique repeat. Black and red represent Z173-S and Z173-R_ZS_, respectively. (**C**) Category of varied abundance of metabolites. (**D**) Number of metabolites increased and decreased in six categories.

**Figure 3 ijms-24-03004-f003:**
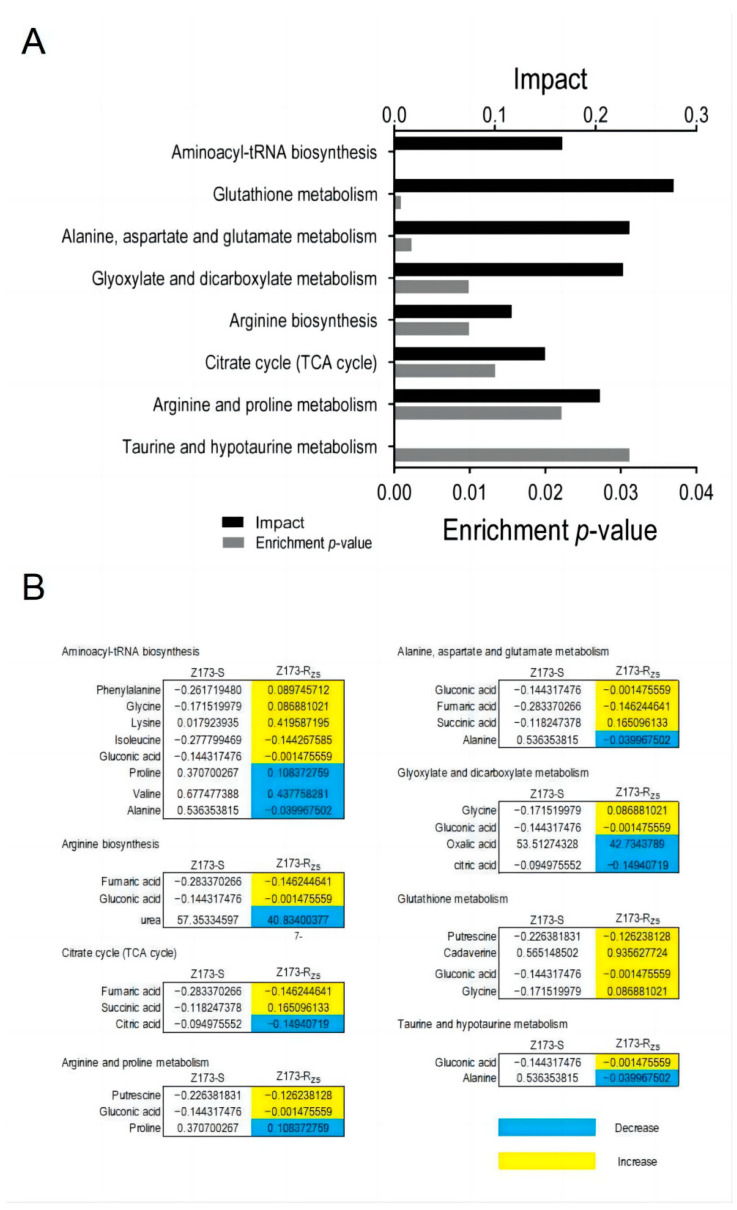
Pathways enrichment. (**A**) Pathways are enriched in differential abundance of metabolites between Z173-S and Z173-R_ZS_. (**B**) Integrative analysis of metabolites of Z173-R_ZS_ compared with Z173-S in significantly enriched pathways. Numbers show the relative values of differential metabolites. Shades of yellow and blue indicate increases and decreases in metabolites, respectively.

**Figure 4 ijms-24-03004-f004:**
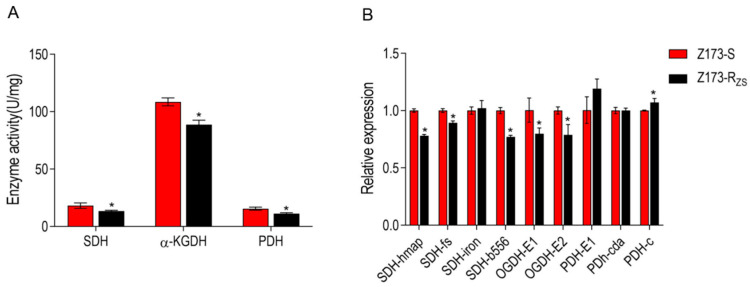
Enzyme activity and gene expression of TCA cycle of Z173-S and Z173-R_ZS_**.** (**A**) Activities of PDH, α-KGDH, and SDH of Z173-S and Z173-R_ZS_ in PSA medium at 30 °C for 18 h. (**B**) qRT-PCR for gene expression of the TCA cycle. Results are displayed as mean ± SD, and significant differences are identified (* *p* < 0.05) as determined by Mann-Whitney U Test. At least three biological repeats were carried out.

**Figure 5 ijms-24-03004-f005:**
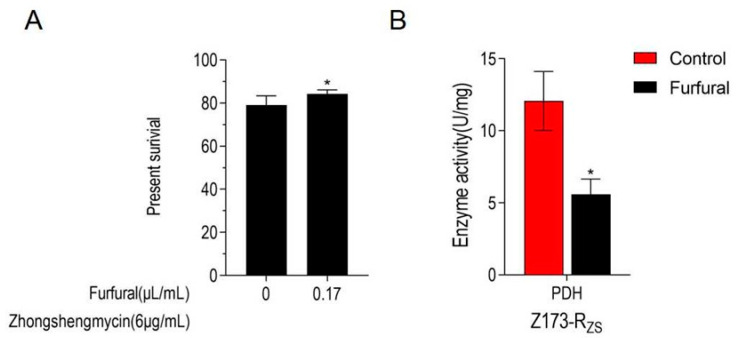
TCA cycle is inhibited in Z173-R_ZS_. (**A**) Percent survival of Z173-R_ZS_ in the presence or absence of inhibitors furfural or malonate. Z173-R_ZS_ was incubated with ZS (6 μg/mL) plus furfural (0, 0.17 μL/mL) in M9 minimal medium, plus NaAc (10.0 mM), MgSO_4_ (2.0 mM), and CaCl_2_ (0.1 mM) at 30 °C for 6 h. (**B**) Activities of PDH in Z173-R_ZS_ in the presence or absence of 0.17 μg/mL furfural. Results are displayed as mean ± SD, and significant differences are identified (* *p* < 0.05) as determined by Mann-Whitney U Test. At least three biological repeats were carried out.

**Figure 6 ijms-24-03004-f006:**
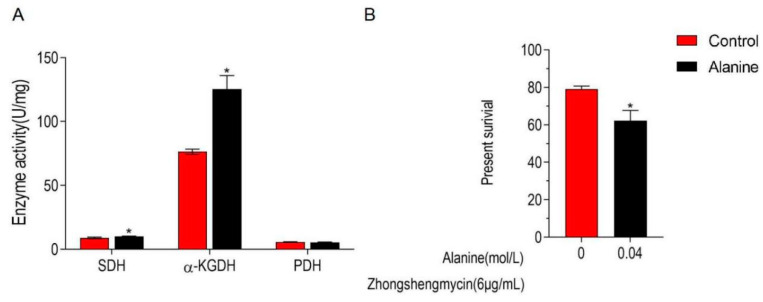
Alanine elevates the TCA cycle and potentiate ZS to control Z173-R_ZS_**.** (**A**) Activities of PDH, α-KGDH, and SDH in Xoo-RZS in the presence or absence of 0.04 M alanine. Results are displayed as mean ± SD, and significant differences are identified (* *p* < 0.05) as determined by Mann-Whitney U Test. At least three biological repeats were carried out. (**B**) Percent survival of Xoo-RZS in the presence or absence of alanine. Xoo-RZS was incubated with ZS (6 μg/mL) plus alanine (0, 0.04 M) in M9 minimal medium, plus NaAc (10.0 mM), MgSO4 (2.0 mM), and CaCl2 (0.1 mM) at 30 ℃ for 6 h.

## Data Availability

Data are contained within the article.

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
