# Peer review of "Alanine-Dependent TCA Cycle Promotion Restores the Zhongshengmycin-Susceptibility in Xanthomonas oryzae"

_ijms, 2023, doi:10.3390/ijms24033004_

Round 1
Reviewer 1 Report
This is a very nice manuscript. It is also has significant scientific merit in terms of the research being presented. There are a few edits and suggests for consideration:
There is inconsistency in the positioning of the reference with sometimes there is a space and sometimes there isn't e.g. vvvv[1] versus vvvv [1]
Microbial antibiotic-resistance, antibiotic resistance should both be referred to as antimicrobial resistance. Antibiotics are the specific drug, whereas antimicrobial is the product specifically and there fore applicable to the general resistance mechanism.
Line 16: delete 'the'
Line 21: replace antibiotics with 'antimicrobial'
When you have numbers in the thousand it should be written as 8,000
Line 122: add a space between 2 and ml
Figure 4, 5, and 6 figure legends are centred and not justified like the others
Reviewer 2 Report
The manuscript (Alanine-dependent TCA cycle Promotion restores the Zhongshengmycin-susceptibility in Xanthomonas oryzae) is interesting but I do suggest the authors discuss in all parts of the manuscript their previous study titled (Exogenous Alanine Reverses the Bacterial Resistance to Zhongshengmycin with the Promotion of the P Cycle in Xanthomonas oryzae) with the current study.
Please explain in detail what is the difference between the two studies as both revealed that alanine was proven to reverse the ZSM resistance in Xoo?
Why did not you combine both studies in a single research as you almost used the same experimental system and materials?
I would like to ask did you use the same strain of the pathogenic bacteria in both studies?
Several expressions are not suitable and common such as killing which is better replaced by control, please carefully check all the manuscript. Revise the English carefully.
L26-28 Long and difficult to understand, please re-write.
L265-268 Please rephrase this sentence in a more understandable way.
Did you measure the growth curve of the isolates?
It is necessary to discuss in detail the outcome of the study results and how can we use the results to control Xoo?
The conclusion is weak please explain the main findings of your study.
Round 2
Reviewer 2 Report
Dear authors
Thank you for answering my questions but I do think that the conclusion (alanine enhances the antimicrobial-mediated Z173-RZS controlling efficiency) is similar to your previous study even if the strain is different.
I do recommend checking for small grammatical errors. Moreover, please add to the conclusion the future studies needed in order to control Xoo.
